# Market Openness and Its Relationship to Connecting Markets Due to COVID-19

Karime Chahuán-Jiménez [1], Rolando Rubilar-Torrealba [2] and Hanns de la Fuente-Mella [3,*]

1 Centro de Investigación en Negocios y Gestión Empresarial, Escuela de Auditoría, Facultad de Ciencias Económicas y Administrativas, Universidad de Valparaíso, Valparaíso 2362735, Chile; karime.chahuan@uv.cl
2 Facultad de Ciencias, Instituto de Estadística, Universidad de Valparaíso, Valparaíso 2360102, Chile; rolando.rubilar@postgrado.uv.cl
3 Escuela de Comercio, Facultad de Ciencias Económicas y Administrativas, Pontificia Universidad Católica de Valparaíso, Valparaíso 2340031, Chile
* Correspondence: hanns.delafuente@pucv.cl

**Abstract:** In this research, statistical models were formulated to study the effect of the health crisis arising from COVID-19 in economic markets. Economic markets experience economic crises irrespective of effects corresponding to financial contagion. This investigation was based on a mixed linear regression model that contains both fixed and random effects for the estimation of parameters and a mixed linear regression model corresponding to the generalisation of a linear model using the incorporation of random deviations and used data on the evolution of the international trade of a group of 42 countries, in order to quantify the effect that COVID-19 has had on their trade relationships and considering the average state of trade relationships before the global pandemic was declared and its subsequent effects. To measure, quantify and model the effect of COVID-19 on trade relationships, three main indicators were used: imports, exports and the sum of imports and exports, using six model specifications for the variation in foreign trade as response variables. The results suggest that trade openness, measured through the trade variable, should be modelled with a mixed model, while imports and exports can be modelled with an ordinary linear regression model. The trade relationship between countries with greater economic openness (using imports and exports as a trade variable) has a higher correlation with the country's health index and its effect on the financial market through its main trading index; the same is true for country risk. However, regarding the association with OECD membership, the relations are only with imports.

**Keywords:** COVID-19; applied econometrics; economic markets; statistical modelling; sustainability



## 1. Introduction

According to Milani [1], with the increase in international financial integration in recent years, bilateral financial links between countries can have an increasing influence on their real economies as well as openness to trade, financial markets, institutions and the sectoral composition of countries and regions. These variables have thus been identified as key determinants of the spatially heterogeneous effects of international financial crises [2]. The evidence suggests that the effects of trade and financial integration do not evolve independently but are complementary in explaining cross-country differences. Nevertheless, the transformative effect of trade (financial) openness decelerates as financial (trade) integration surpasses the estimated optimal levels [3]. The financial, economic and health crises we are now confronting have reignited the fierce debate on the merits of financial globalisation and its implications for growth, particularly for developing countries [4].

Following the recent global financial crisis, suitable models for modelling the non-normal events leading to these crises have become essential in the fields of finance and risk management, among which the volatility series are a good indicator of investor expectations and perceived risk [5]. With the devastating effects of COVID-19 in all spheres of human

life, reflected in the number of people infected by and having died due to COVID-19, countries have responded by restricting economic activity and peoples' mobility, imposing travel bans and implementing stimulus packages to cushion the unprecedented slowdown in economic activity and loss of jobs; as is the case with any unexpected news, markets overreact and as more information becomes available and people more broadly understand the ramifications, the market corrects itself [6]. When governments overreact, these effects will be passed on to the market. We should therefore see an overreaction from the market and, as governments correct their reactions, the market will also do so [6]. This implies that global financial markets reflect COVID-19, and could be affected by regional decisions and then by a contagion effect at the global level depending on country conditions.

In addition, according to Moon, Mun and Lee [7], the best performance in terms of a country's total output results from the regime under which capital and labour markets are open.

Financial markets experience crises differently, irrespective of whether these effects are contagious [8]. In the case of emerging countries, according to Gupta, Mishra and Sahay [9], crises are one and a half times more likely to be contractionary in emerging markets than in other developing economies. The enforcing of strict actions and policies affecting globalisation and trade were necessary for controlling the pandemic [10]; the variable number of COVID-19 cases per million inhabitants is statistically significant, showing its impact on each country's economy through the GDP variation. Therefore, there have been reports of how COVID-19 cases affect domestic economies, in addition to other relevant risk factors such as OECD membership [11]. Thus, the effects on global markets can to a greater extent be related to certain variables considered in the literature that have not been related in previous studies, such as trade openness (exports and imports), COVID-19 and country development.

The investigation by Ahmar and del Val [12] and the International Monetary Fund in 2018 indicated that global growth for 2017 had strengthened by 3.8% and trade had increased worldwide. Considering the projections of a global growth increase of 3.9% between 2018 and 2019, this growth was especially driven by developing markets and the expansion of developed countries, before the economic growth started to fall in 2020 [13]. Economic intuition suggests that when two economies are well integrated through trade, investment and financial relationships [14], a crisis in one economy is likely to spread quickly to the other, as exposure to financial globalisation can lead to the increased vulnerability of a financial situation [15]. The results show that the COVID-19 pandemic in China had a significant negative effect on its export trade; that the COVID-19 pandemic situations in the trading partner countries and regions generated significant positive effects for China's total exports; and that the COVID-19 pandemic thus had a heterogeneous impact on China's exports towards different trading partners [16]. Thus, the interactive effect was significant in the Chinese stock market, exacerbating abnormal market volatilities and risk contagion [17].

Global trade is one mechanism through which geographically distant epidemics may affect countries unaffected by disease. In terms of infectious diseases, scholars have discovered that epidemics can impact short-term economic performance by changing expectations and deterring investment [18]. Some studies showed that the impact of COVID-19 had been reflected in the return of financial markets, the volatility of financial markets, and the risk contagion among financial markets [19,20]. Fu [21] showed that, using extreme dependency contagion tests, contagion effects are widespread in global stock markets in four regions: Latin America and North America are highly exposed to contagion risks, followed by Europe, and Asia being the least vulnerable. However, a link between epidemics and long-term growth has not been established [18]. A study by Kostova et al. [18] indicated that US exports and the jobs created by these exports were negatively affected by the 2014 Ebola outbreak in West Africa, and international trade rapidly collapsed during the 2020 global recession [22].

Tan and You-Hung [23] showed that extreme events have negative and multidimensional impacts on economic and social health, particularly at the macroeconomic level.

For Keogh-Brown, Smith, Edmunds and Beutels [24], there is no econometric method suitable for estimating the likely cost of a pandemic, the benefits of policies to mitigate the effects of disease, or the distribution of the costs and benefits of a disease within an economy. For Kostova et al. [18], the immediate economic costs of epidemics in affected regions can be substantial and are thus frequently assessed following outbreaks, as well as the economic responses implemented by governments during pandemics such as income support, fiscal measures and international aid, which all have a restrictive effect—especially on exchange rate volatility [25].

The new macroeconomic theory of the open economy incorporates imperfect competition and nominal rigidities in a dynamic general equilibrium [26], such as excessive exchange rate movements and liquidity effects that were not previously accounted for [27–30]. Kumar [31] examined the bilateral trade relationship in terms of world trade share, bilateral trade flow, trade intensities, trade reciprocities and indices such as the export intensity index, import intensity index and trade reciprocity index, and revealed the comparative advantage index in order to analyse the trade relationship.

For Milani and Park [32], globalisation has introduced important changes in the macroeconomic environment. Domestic variables have become much more sensitive to global measures; in particular, domestic output and inflation are significantly affected by global output. For Tiryaki [33], the conclusion of their investigation was that the price of imported inputs and the productivity of the non-tradable sector are the two most important sources of macroeconomic fluctuations in a typical emerging market economy. Interest rates and the price of imported final products also play an important role in driving investments and import fluctuations.

The main macroeconomic variables, according to Popper, Mandilaras and Bird [34], are associated with three combined options for a country, linked to exchange rate setting, financial openness and monetary sovereignty.

According to Kahiya [35], the export variable is defined as selling products and/or services via a direct and/or indirect method in the foreign market from a firm's production facilities in its country. In addition, this variable, according to Alexandre, Costa, Portela and Rodriguez [2], is an opportunity to diversify markets, which is associated with firms that are more innovative companies and have higher productivity.

For Clark and Kassimatis [36], a higher level of output means a higher debt-servicing capacity of the economy; many studies find that gross domestic product (GDP) growth or a similar activity-based indicator is a significant deterrent to sovereign spreads, and its terms (price of exports relative to the price of imports) is another important determinant of sovereign spreads, as it affects the economy's ability to generate the current foreign exchange earnings necessary to service external debt. An empirical examination of the aforementioned relationship finds that the terms of trade, as well as the volatility of trade rates, are significant factors affecting the sovereign spreads of emerging economies [36]; they also report that the terms of trade have a significant inverse relationship with sovereign spreads.

Tebaldi, Nguyen and Zuluaga [37] showed that in emerging economies, from 1994 to 2014, GDP growth, the real effective exchange rate and political liberalisation play key roles in determining differentials. Milani and Park [1] indicated that fluctuations in foreign stock prices continue to influence domestic expectations of future output gaps in all countries.

Therefore, the research presented herein links the variables present in the literature that have not previously been related, which are associated to a global pandemic on the one hand, and on the other, are variables that affect global markets. From the point of view of the pandemic, the variables that are considered relevant are the health index of a country, and from the point of view of the effects on the global market by financial contagion, the relevant variables are the country risk, GDP, economic openness (imports and exports) and the development of the country.

As such, this research will contribute to addressing the problem from an econometric point of view, as through the analysis of the current pandemic, we will determine how soon the global economic is likely to recover and help in preparing for future outbreaks.

While some work has been done on how to mitigate the spread of the pandemic and its implication for the global economy [38–40], there is still much to be desired in terms of overall sustainability.

The present investigation presents the following hypothesis: the macroeconomic indicators have further deteriorated in countries with greater trade openness. Studies of the effect that the global pandemic (COVID-19) has had on the economy, considering the trade openness of countries as a key element, are characterised according to Table 1.

**Table 1.** Variables used in the literature in relation to the health crisis.

| Concept | Variable | Reference |
|---|---|---|
| Financial Integration:<br>- Bilateral Financial connection<br>- Trade Opening<br>- Financial Markets | Trade | Milani [1]<br>Ibrahim [3]<br>Kose et al. [4]<br>Gupta et al. [9]<br>Pan and Yue [10]<br>Phan and Narayan [6]<br>Ivanov et al. [14]<br>Chahuán-Jiménez et al. [19]<br>Wu and Hui [41]<br>Kumar [31]<br>Popper et al. [34] |
| Export and Import | Export and Import | Moon et al. [7]<br>Zhao et al. [16]<br>Clark and Kassimatis [36] |
| Infected and Dead,<br>For Economic Activity<br>and GDP | Health<br>Cases per Million People<br>GDP | Phan and Narayan [6]<br>de la Fuente-Mella et al. [11]<br>Albu et al. [13]<br>Zhao et al. [16]<br>Kostova et al. [18]<br>Benguria [22]<br>Ibrahim [3]<br>de la Fuente-Mella et al. [11]<br>Clark and Kassimatis [36] |
| Volatility (Risk) | Risk / SDRisk | Lahmiri and Bekiros [5]<br>Feng et al. [25] |
| Production and Open Markets | GDP | Moon et al. [7]<br>Tan and Yu-Hung [23]<br>Milani and Park [32] |
| Emerging economies<br>(OECD and non-OECD) | OCDE | Tebaldi et al. [37]<br>Tiryaki [33] |

The method applied in this research was based on a mixed linear regression model that contains both fixed and random effects for the estimation of parameters and a mixed linear regression model corresponding to a generalisation of a linear model using the incorporation of random deviations.

The analysis will involve three types of measurements: first, trade openness as the sum of imports and exports; second, trade openness measured only through imports; and third, trade openness measured only through exports—thus generating three independent models for each of the cases, through a mixed linear regression model. Additionally, for each of the cases, the variables of health, GDP, country risk and OECD membership will be considered.

The results suggest that trade openness, measured through the trade variable, should be modelled with a mixed model, while imports and exports can be modelled with an ordinary linear regression model. Analysing the models presented in this investigation, in the model associated with trade and imports, the independent variables are significant, while they are not significant for exports. This can be explained by the fact that countries

tend to primarily limit their foreign trade relationships by controlling imports while attempting to keep the revenue or export channel open. This phenomenon suggests that in the event of border closures, there is a degree of import substitution, favouring local industry, a phenomenon that was not studied in this paper but that emerges as a hypothesis for future investigations in addition to other lines of research, such as the analysis of the herding behaviour phenomenon and the sentiment analysis in financial markets when facing global crises such as COVID-19.

This paper is organised as follows. Section 1 presents the introduction which includes a literature review, and the materials and methods are presented in Section 2. The results are shown in Sections 3 and 4 corresponds to the discussion, conclusions, limitations, and future research.

## 2. Materials and Methods

This research used data on the evolution of the international trade of a group of 42 countries, as listed in Table 2, in order to quantify the effect of COVID-19 on their trade relationships, considering the average state of trade relationships before the global pandemic was declared and its subsequent effects. The list of countries used in this investigation considered two specific groups of countries—developed countries and developing countries—with the purpose of capturing the differential effects that these groups may have on international trade, for which information was obtained from available databases, as shown below.

**Table 2.** Countries used in this study.

| Country | OECD | Country | OECD | Country | OECD |
|---|---|---|---|---|---|
| Australia | Yes | Israel | Yes | Romania | No |
| Belgium | Yes | Japan | Yes | Russia | No |
| Brazil | No | Korea | Yes | Saudi Arabia | No |
| Bulgaria | No | Malaysia | No | Serbia | No |
| Canada | Yes | Mexico | Yes | Slovakia | No |
| Chile | Yes | New Zealand | Yes | Spain | Yes |
| China | No | Nigeria | No | Sri Lanka | No |
| Colombia | Yes | Norway | Yes | Sweden | Yes |
| Cyprus | No | Pakistan | No | Switzerland | Yes |
| Egypt | No | Peru | No | Thailand | No |
| Finland | Yes | Philippines | No | Turkey | Yes |
| Hungary | Yes | Poland | Yes | Ukraine | No |
| Iceland | Yes | Portugal | Yes | UK | Yes |
| India | No | Qatar | No | USA | Yes |

By measuring the percent variation in trade relationships, imports, exports and the sum of both, it is possible to indicate the effects on the reactive capacity of countries induced by unexpected global health shocks, as in the case of the COVID-19 health crisis.

To measure, quantify and model the effect of COVID-19 on trade relationships, three main indicators were used, i.e., imports, exports and the sum of imports and exports, which are calculated by averaging the first and second quarters of 2016, 2017, 2018 and 2019, compared to the average of the first and second quarters of 2020. Only the first two quarters of each year were used to prevent a seasonal effect of the different countries' trade relationships, assuming that the average of the last 4 years represents the general state of the trade relationships of the group of countries analysed in this investigation.

With these measurements, a mixed linear regression model was proposed that contains both fixed effects and random effects for parameter estimation. A mixed linear regression model corresponds to the generalisation of a linear model by incorporating random deviations that may be associated with the error term.

The functional relationship of the model is defined as follows:

$$Y_{ij} = f(x_{1i}, x_{2i}, x_{3i}, x_{4i}, x_{5i}, x_{6i}, x_{7i}, x_{8i}, x_{9i}, x_{10i}, x_{11i}, u_j),$$

(1)

where $Y_{ij}$ corresponds to the effect of COVID-19 on the trade relationship of the countries measured as the percent change between the pre-pandemic and post-pandemic trade relationship in each country $j$ or each trade relationship $i$ [42]; $x_{1i} = Health_i$, which corresponds to the health index of the country $i$ as measured by the global health security (GHS) index for 2019 [43]; $x_{2i} = Cases_i$, which corresponds to the number of COVID-19 patients' rate measured by the number of COVID-19 cases at the peak of the pandemic per million inhabitants until 31 December 2020 [44]; $x_{3i} = SqCases_i$, which corresponds to the square of *Cases* to capture the non-linearity of the variable; $x_{4i} = \log(Risk_i)$, which corresponds to the natural logarithm of the average risk of country $i$ in 2019; and $x_{5i} = \log(StdRisk_i)$, which corresponds to the natural logarithm of the standard deviation of the risk of country i in 2019. Additionally, seven control variables were incorporated into the model: $x_{6i} = OECD_i$, which takes a value of 1 if the country belongs to the OECD group of countries and 0 if it does not; $x_{7i} = \log(GDP_i)$, which corresponds to the natural logarithm of the GDP of country $i$ in 2019 [45]; and the variables $x_{8i} = America_i$, $x_{9i} = Asia_i$, $x_{10i} = Europe_i$ and $x_{11i} = Oceania_i$, which takes a value of 1 if the observation of the country's relationship corresponds to trade with the respective continent.

Accordingly, in the model proposed for the present investigation, the health variable is an indicator of each country's level of development in terms of its health response, considering different scenarios, and is thus representative of the countries' response capacity to health emergencies at the time prior to the formal declaration of the existence of a global pandemic. On the other hand, the variable of the number of COVID-19 patients rate represents the direct impact that the pandemic has had on each of the countries in the sample.

In turn, the country risk variables (risk and StdRisk) are measured by the CDS on government bonds from execution to five years, as provided by the Bloomberg database, and represent an assessment of a country's level of risk vis-à-vis external investment at the time prior to the formal declaration of the existence of a global pandemic.

The OECD membership variable makes it possible to identify the level of development of each country and the effects it may have on international trade. On the other hand, the GDP enables one to know the size of the market and its relevance in the international markets.

The variables of the model are shown in Figure 1, where the upper part shows the correlation between the main variables considered in the model, where trade, import and export correspond to the independent variables of the model. There is a strong correlation between the Trade and Import variables with respect to GDP, suggesting their incorporation within the regression model. There is also a strong correlation between the health, risk and StdRisk variables, which could imply problems of multicollinearity.

In addition, the scatter plots between the variables under study are shown at the bottom of the graph, while on the diagonal, we can see the respective histogram of each variable, allowing a graphical analysis of the variables under study.

To estimate the regression model based on Equation (1), a mixed model was established to explain the response variable Y (export, import and export plus import) in terms of the explanatory variables [46,47]. The specific mixed model assumed in this investigation is as follows:

$$Y_{ij} = \beta_0 + \sum_{s=1}^{S} \beta_s x_{si} + u_{0j} + \varepsilon_{ij}, \tag{2}$$

where $\beta_0$ corresponds to a parameter associated with the model constant; the parameters $\beta_1, \beta_2, \beta_3, \beta_4, \beta_5, \beta_6, \beta_7, \beta_8, \beta_9, \beta_{10}, \beta_{11}$ correspond to the parameters of the linear model associated with the variables.

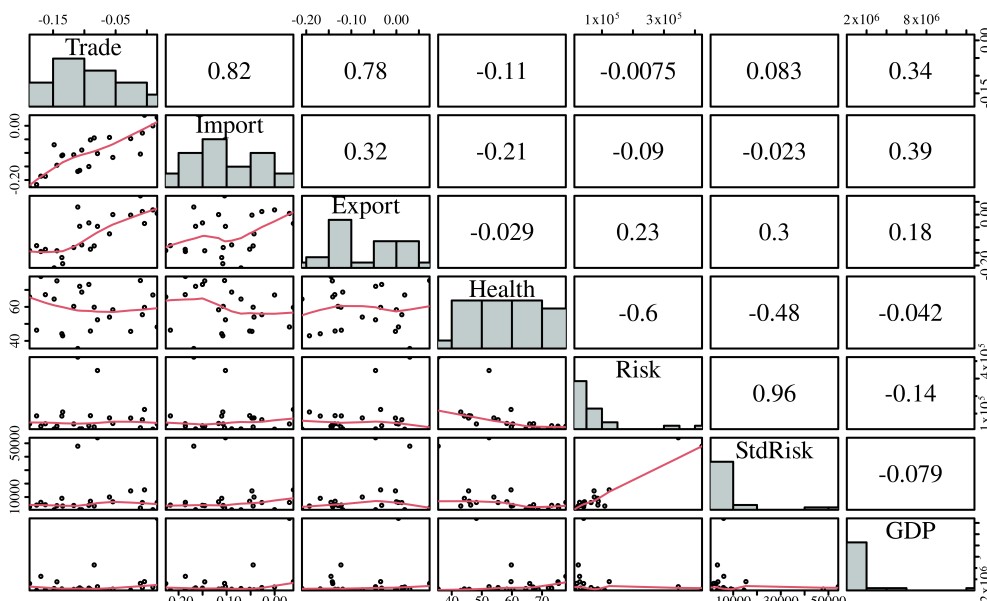

**Figure 1.** Histograms, Pearson correlations and scatterplots of the variables included in the model.

The random factor of the mixed model $(u_{0j} + \varepsilon_{ij})$ assumes that $u_{0j}$ is orthogonal to $\varepsilon_{ij}$. $u_{0j}$ corresponds to the random intercept, representing a change in the overall average $\beta_0$, which is associated with the aggregate trade of each country $i$.

The structure of the variance is as follows:

$$Var\begin{pmatrix} u \\ \varepsilon \end{pmatrix} = \begin{pmatrix} G & 0 \\ 0 & \sigma_\varepsilon^2 R \end{pmatrix}, \tag{3}$$

where $G$ corresponds to the variance and covariance matrix of $u$, and $\varepsilon$ corresponds to a multivariate normal matrix with a mean of 0 and a variance pf $\sigma_\varepsilon^2 R$. In the estimation procedure, $u$ is not directly calculated but is characterised by the elements of $G$, known as variance components, which are estimated from the average variance in the residual $\sigma_\varepsilon^2$, and the variance in the residual is contained within $R$.

In principle, we were not interested in the trade evolution of a particular country but rather in the variability of a sample of countries in response to the phenomenon of the COVID-19 pandemic—and the country is therefore considered as a random effect.

The fixed part of the model represents the average behaviour of the countries' trade relationships, while the random part represents a change in the slope for each country.

### 3. Results

This section presents six model specifications for the variation in the foreign trade response variable. In the above, models of imports, exports and the sum of imports and exports (trade) are considered and given the following explanatory variables for the following specifications: (i) health index; (ii) number of patients rate; (iii) square of number of patients rate; (iv) logarithm of the average country risk; (v) logarithm of the standard deviation of country risk; (vi) membership or non-membership in the OECD group of countries; and (viii–xi) whether the trade relationship is associated with the Americas, Asia, Europe or Oceania, respectively. In the case of specifications 4–6, the following explanatory variables are considered: (i) health index; (ii) number of patients rate; (iii) square of the number of patients rate; (iv) logarithm of the average country risk; (v) logarithm of the standard deviation of country risk; (vi) membership or non-membership in the OECD group of countries; (vii) logarithm of GDP; and (viii–xi) whether the trade relationship is associated with the Americas, Asia, Europe or Oceania, respectively.

Table 3 provides the results of the regression analysis for the models of the variation in the international trade indicators. For specifications 1–3, the health variable associated with the health level of the countries under study is non-significant for the three models. The Cases and SqCases variables are significant at 5% for the case of the models of trade and exports. For the log(Risk) variable, we note that it is significant for the imports model, as well as the log(StdRisk). The OECD variables are significant for the trade and import model. All the variables associated with trade with a particular continent are non-significant except for Europe in the trade model.

For specifications 4–6, the health variable is only significant for the export model. Cases and SqCases variables are significant for the trade and export models, as log(Risk) and log(StdRisk) variables are significant for the trade and import models, while the OECD and log(GDP) variables are only significant for the import model.

**Table 3.** Parameter estimate and the corresponding standard error (in parenthesis) of the indicated model, as well as the statistical indicators of goodness-of-fit and significance.

| | Trade | Import | Export | Trade | Import | Export |
|---|---|---|---|---|---|---|
| Health | 0.000524 | −0.00189 | 0.00293 | 0.00191 | −0.000817 | 0.00426 * |
| | (0.00227) | (0.00298) | (0.00231) | (0.00234) | (0.00314) | (0.00239) |
| Cases $\times 10^3$ | 0.290 ** | 0.0235 | 0.413 *** | 0.298 ** | 0.0299 | 0.421 *** |
| | (0.133) | (0.175) | (0.135) | (0.128) | (0.173) | (0.131) |
| SqCases $\times 10^3$ | −0.0219 ** | −0.00411 | −0.0282 *** | −0.0216 ** | −0.00392 | −0.0279 *** |
| | (0.0103) | (0.0136) | (0.0105) | (0.0100) | (0.0135) | (0.0102) |
| log(Risk) | −0.0154 | −0.130 ** | 0.0650 | −0.0333 | −0.144 ** | 0.0479 |
| | (0.0469) | (0.0618) | (0.0478) | (0.0465) | (0.0627) | (0.0476) |
| log(SDRisk) | −0.000315 | 0.0807 * | −0.0482 | 0.0172 | 0.0943* | −0.0314 |
| | (0.0357) | (0.0470) | (0.0364) | (0.0360) | (0.0485) | (0.0368) |
| OECD | −0.0831 * | −0.127 ** | −0.0504 | −0.0832 ** | −0.127 ** | −0.0505 |
| | (0.0427) | (0.0562) | (0.0435) | (0.0413) | (0.0555) | (0.0422) |
| log(GDP) | | | | −0.0237 * | −0.0184 | −0.0228 |
| | | | | (0.0138) | (0.0186) | (0.0142) |
| Continent | | | | | | |
| America | 0.000112 | 0.0269 | −0.0182 | 0.000112 | 0.0269 | −0.0182 |
| | (0.0412) | (0.0627) | (0.0462) | (0.0412) | (0.0627) | (0.0462) |
| Asia | 0.00110 | 0.0131 | 0.0105 | 0.00110 | 0.0131 | 0.0105 |
| | (0.0412) | (0.0627) | (0.0462) | (0.0412) | (0.0627) | (0.0462) |
| Europa | −0.0712 * | −0.0845 | −0.0621 | −0.0712 * | −0.0845 | -0.0621 |
| | (0.0412) | (0.0627) | (0.0462) | (0.0412) | (0.0627) | (0.0462) |
| Oceania | −0.0211 | −0.0590 | 0.0127 | −0.0211 | −0.0590 | 0.0127 |
| | (0.0412) | (0.0627) | (0.0462) | (0.0412) | (0.0627) | (0.0462) |
| Constant | 0.0815 | 0.862 | −0.571 | 0.353 | 1.073 * | −0.310 |
| | (0.417) | (0.549) | (0.425) | (0.433) | (0.584) | (0.444) |
| log() Var(Constant) | −2.638 *** | −2.676 *** | −2.792 *** | −2.723 *** | −2.730 *** | −2.901 *** |
| | (0.272) | (0.525) | (0.392) | (0.303) | (0.573) | (0.462) |
| log() Var(Residual) | −1.667 *** | −1.248 *** | −1.553 *** | −1.667 *** | −1.248 *** | −1.553 *** |
| | (0.0546) | (0.0546) | (0.0546) | (0.0546) | (0.0546) | (0.0546) |
| LR Test vs. Linear Model | | | | | | |
| $\chi^2(1)$ | 5.45 | 1.12 | 2.20 | 4.12 | 0.92 | 1.50 |
| $Prob > \chi^2$ | 0.01 | 0.14 | 0.07 | 0.02 | 0.16 | 0.11 |
| N | 210 | 210 | 210 | 210 | 210 | 210 |

Standard errors in parentheses * $p < 0.10$, ** $p < 0.05$, *** $p < 0.01$.

We then checked whether the specification of the models' corresponds to an ordinary linear regression model or a mixed model. For this, we used the likelihood ratio test, which allows testing the goodness of fit between two models, in our case, a restricted model which corresponds to the ordinary linear regression model, and the mixed model.

Rejection of the null hypothesis implies that there is a significant difference in the use of one model or the other. For models 1–3, the null hypothesis is rejected only for the trade indicator at a significance level of 5%. Conversely, the null hypothesis is not rejected for the import and export models, and thus, modelling with an ordinary linear regression model is suggested. For specifications 4–6, findings similar to those for specifications 3–6 are observed, where only the trade model is significant, and there is evidence that the best model is a mixed model.

To measure the possible effects of multicollinearity, the variance inflation factors ($VIF$) of models 1–6 are measured. For models 1–3, we observe that: $VIF(Health) = 2.79$; $VIF(Cases) = 9.07$; $VIF(SqCases) = 8.65$; $VIF(log(Risk)) = 7.58$; $VIF(log(SDRisk)) = 5.84$; $VIF(OECD) = 1.56$; similar values are evidenced for models 4–6, adding the $VIF(log(GDP)) = 1.47$. The average VIF is 4.19 for models 1–3 and the average VIF is 4.06 for models 4–6 . These results suggest that there are no serious multicollinearity problems.

The results suggest that trade openness, measured through the trade variable, should be modelled with a mixed model, while imports and exports can be modelled with an ordinary linear regression model.

## 4. Discussion, Conclusions, Limitations, and Future Research

Financial markets experience economic crises irrespective of the effects corresponding to financial contagion [8]. For Chahuán et al. and Alqaralleh and Canepa [19,20], COVID-19 caused a structural failure based on a financial contagion, influenced by the variables of health, country risk measured by the CDS, OECD membership and GDP, variables that were proposed by [1,2,36,37]. Our research expands the observed effects of the COVID-19 pandemic, showing its relevant impacts on international trade, causing problems in the import and export processes of countries at the global level.

Our research shows the effects on countries at the global level and in line with [3,4,9], countries with greater economic openness (imports and exports as a trade variables) have a higher correlation between their health index and the effect on the financial market through the main trading index—the same is true for country risk. However, regarding the association with OECD membership, the link is only with imports, and log GDP is also linked to imports, as in the previous case.

The proposed models have three groups of variables that affect the phenomenon of international trade. The first group of variables corresponds to health characteristics, captured by the health emergency response capacity (health) and the specific effect that the COVID-19 pandemic has had in each country in the sample (cases and SqCases). The health variables show that the COVID-19 pandemic has had a direct effect on exports, suggesting that the effect was reflected in the decreased productive capacity of each country in the sample.

The second group of independent variables corresponds to variables associated with financial risk (log(Risk) and log(SDRisk)), which show a direct effect on imports. The effect on imports from the countries of the sample shows us that the level of risk of the countries can affect the import capacity due to external restrictions; in other words, the countries that have seen their production capacity affected will export to countries with a lower level of risk.

The third group of variables associated with the level of development of the countries in the sample (OECD and log(GDP) ) is significant for trade openness (trade). The above shows that the countries with the highest level of development carried out a greater contraction of their international trade, which suggests that they took more aggressive measures to control the dissemination of COVID-19, adversely affecting the trade relations of these countries.

From the analysis of the models presented in this investigation, we judge that the independent variables are significant in the model associated with trade and imports, though they are not significant for exports. This can be explained by the fact that countries primarily tend to limit their foreign trade relationships by controlling imports while at-

tempting to keep the revenue or export channel open. This phenomenon suggests that in the event of border closures, there is a degree of import substitution, favouring local industry, a phenomenon that was not studied in this paper but that emerges as a hypothesis for future investigations.

There is a link between epidemics and short-term economic performance [18,19]. The literature has identified that global health problems have direct and immediate impacts on both financial markets and the real economies of individual countries. Variables such as health level, country risk, development level and size (GDP) seem to be the main determinants of this relationship. However, these short-term relationships can be expanded by the interaction between countries coming from their trade relations, causing repercussions that can be significant in the medium or long term.

Consequently, the trade relationships between countries, as measured by the sum of imports and exports, and in particular by imports, are significant considering that the countries under study—confronted by a market shock caused by a pandemic—seek alternatives to imports in response to market closures and diversify their exports. Country risk, which is significantly reflected in the present investigation, indicates that the closure of countries' economies makes it possible to conclude that the size of a country's economy is influential.

This investigation generates future research lines such as the analysis of the herding behaviour phenomenon and sentiment analysis in financial markets when facing global crises such as COVID-19 or another global crisis in the markets, which makes it possible to understand the behaviour of market operators and to provide information to policy makers on the effects that other global crises may have.

**Author Contributions:** Data curation, H.d.l.F.-M., K.C.-J. and R.R.-T.; formal analysis H.d.l.F.-M., K.C.-J. and R.R.-T.; investigation, H.d.l.F.-M., K.C.-J. and R.R.-T.; methodology, H.d.l.F.-M., K.C.-J. and R.R.-T.; writing—original draft, H.d.l.F.-M., K.C.-J. and R.R.-T.; writing—review and editing, H.d.l.F.-M., K.C.-J. and R.R.-T. All authors have read and agreed to the published version of the manuscript.

**Funding:** Research work of H. de la Fuente-Mella was partially supported by a grant from the Núcleo de Investigación on Data Analytics/VRIEA/PUCV/039.432/2020 from the Vice-Rectory for Research and Advanced Studies of the Pontifical Catholic University of Valparaíso, Chile.

**Institutional Review Board Statement:** Not applicable.

**Informed Consent Statement:** Not applicable.

**Data Availability Statement:** The data used to support the findings of this study are available from the corresponding author upon request.

**Conflicts of Interest:** The authors declare no conflict of interest.

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
