# Peer review of "Market Openness and Its Relationship to Connecting Markets Due to COVID-19"

_sustainability, doi:10.3390/su131910964_

Round 1
Reviewer 1 Report
- It is necessary to reconsider whether the selection of samples from 24 countries is appropriate. Of the 24 countries, 17 OECD member countries and 7 non-OECD countries, the reason why the balance is not right should be explained.
- There is a debate that the GHS index used as a health index is inappropriate to be used as a ranking. Therefore, it is recommended to use the number of corona19 confirmed case relative to the population.
- The reason why imports have a higher correlation with GDP than exports is thought to require a logical explanation rather than statistics.
Reviewer 2 Report
See attached file.

Reviewer 3 Report
The problems investigated in the paper are interesting. However, the paper has a few serious drawbacks which must be corrected.
- The title does not reflect the content. The functioning of financial markets and their possible failure are not analysed in the paper. The title must be changed.
- The abstract does not clearly state the main aims and methodology employed as well as the results. The description is rather muddled and should be improved.
- The introduction should be reorganised. Now it is a little bit chaotic and does not point the main field of interest of the authors.
- At the end of the introduction, the structure of the paper should be described.
- There is no appropriate justification why the analysed countries were chosen.
- The sample is rather small what could undermine the results of the paper.
- A more exhausting description of variables employed is necessary. Moreover, the authors use the GDP as X5 – it is not clear whether it is global GDP, GDP per capita, % change in GDP, or logarithm of GDP.
- There are unnecessary repetitions in part 3. Results (description of variables etc.).
- No hypotheses are proposed.
- It is not clear what a “simple regression model” means.
- The discussion is too short. There should be a more extensive discussion of the results obtained in comparison with the results described in the literature.
- Additional literature items should be added.
Reviewer 4 Report
The article is very interesting and addresses an important issue in the context of COVID19. The objective of the paper is clear and the econometric methodology is rigorous.
However, there are some minor issues that need to be addressed in order to improve the paper.
1. Although the introduction is very interesting and the included literature has been reviewed, the gaps in the literature and the contribution of this article to fill these gaps are missing.
2. With regard to the contagion of international markets, I suggest that the authors look for recent evidence to strengthen their argument on the originality of their study
4. The empirical part is very well elaborated and methodical. The results are well presented.
3. It seems to me that the discussion can be improved. I would like the author(s) to elaborate and explain the novelty of their analysis and compare the results with the literature.
Round 2
Reviewer 3 Report
I do not have any further comments.
Author Response
Thank you very much for your comments.